# LAM Simulator: Advancing Large Action Model Training for Agent via Online Exploration and Feedback Simulation

## Abstract

Large Action Models (LAMs) for AI agents have significant potential, but their development is often constrained by the reliance on supervised learning and manual data curation, which are both time-consuming and costly. To address these limitations, we present the LAM Simulator, a comprehensive framework designed for online exploration of agentic tasks with high-quality feedback. This framework includes a curated set of high-quality agentic tasks, a diverse collection of tools, and an interactive environment where agent models can call tools, receive execution responses, and obtain action feedback. Our findings indicate that the LAM Simulator significantly enhances model performance and effectively identifies and addresses potential issues. Specifically, our model, LAM-Sim-8x7B, demonstrates an 18.54% improvement over its base LAM and significantly outperforms other state-of-the-art alternatives on ToolEval benchmark. Furthermore, we have demonstrated that LLMs lacking in agentic capability can greatly benefit from the implementation of LAM Simulator. Our experiments with a model trained on Mixtral-8x7B-Instruct-v0.1 have yielded a doubling to tripling of performance. Remarkably, the data construction process for training these models requires minimal human intervention, making the LAM Simulator a robust framework for accelerating the development of AI agents.

## 1 Introduction

Large Action Models (LAMs) (Zeng et al., 2023; Xu et al., 2024; Liu et al., 2024b; Zhang et al., 2024b) are an advanced type of Large Language Model, specifically optimized for tool usage, reasoning, and function calling. Recent advancements have propelled their capabilities, making them integral to applications such as smart assistants, AI agents, and task automation. While strong closed-world commercial models like Claude-3 (Anthropic, 2024) and GPT-4 (Achiam et al., 2023) can also perform complex agent tasks, LAMs benefit from specialized training for enhanced performance in agent applications and offer more open-source options and developments for the community. As use cases grow, the demand for more accurate models will continue to increase.

Current approaches for creating open-sourced LAMs include prompt engineering, incorporating additional contextual information into agent prompts, Supervised Fine-Tuning (SFT), Reinforcement Learning from Human Feedback (RLHF) (Ouyang et al. (2022)), among others. However, most of these methods rely heavily on supervised learning and manual data curation, a process that is both time-consuming and expensive.

The concept of agent self-exploration has emerged as a promising avenue for reducing human labeling and annotation effort in the development of agent models. Recent studies, including those by ToolTalk (Farn & Shin, 2023), WebArena (Zhou et al., 2023), and APIGen (Liu et al., 2024b), have demonstrated the ability to generate high-quality data for agent learning and evaluation through automated means. However, these still have some limitations. ToolTalk is limited to tasks that are curated or filtered by humans. WebArena is constrained by a specific set of tasks and very limited action spaces within the web environment domain. APIGen, while showing considerable potential in generating function-calling data, is limited to single-turn function calling and primarily focused on ensuring the correctness of function names and corresponding parameters.

On the other hand, existing open-source agent models such as Lemur (Xu et al., 2024), AgentLM (Zeng et al., 2023), and the latest xLAM (Zhang et al., 2024b) still rely on rule-based methods and strong LLMs for collecting and filtering agent trajectories. Moreover, without a careful process of providing feedback or crafting the dataset, it is difficult to resolve the agent's issue of handling specific errors such as JSON parsing errors, tool hallucinations, and argument inaccuracies. This approach also limits the agent model's ability to explore a broader range of states based on past agent trajectories and to identify potential improvements.

In light of these limitations, we present the LAM Simulator, an all-encompassing framework for AI agent learning. Our framework enables AI agent models to interact with environments and tools in real-time, facilitating problem-solving and providing feedback useful for model learning at any point in the process, including both action-wise feedback and task-wise feedback. This enables a wide ranges of applications, from generating pairwise data for Direct Preference Optimization (DPO) (Rafailov et al., 2024) training, to generating high-quality data for supervised training. Figure 1 illustrates our framework.

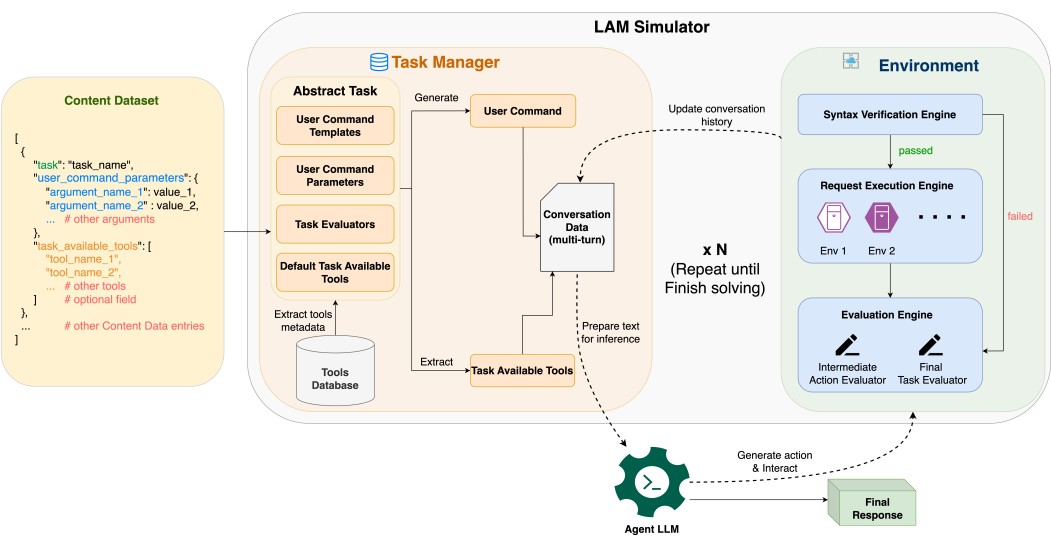

Figure 1: Overview of the LAM Simulator. This figure illustrates the framework's components and their interactions, highlighting the simulator's capabilities in generating tool-use data, executing functions, and evaluating results.

Our experiments demonstrate the remarkable effectiveness of the LAM Simulator in improving model performance and identifying and addressing model weaknesses in an automated manner.

## 2 RELATED WORK.

With the rapid evolution of Large Language Models (LLMs), there has been a significant increase in their application to tool-use and function-calling scenarios. Enhancing the capabilities of LLMs (Achiam et al., 2023; Anthropic, 2024; Dubey et al., 2024; Zhang et al., 2024b) with external tools allows them to go beyond the limitations of their static parametric knowledge and text-based input-output interfaces. This extension enables them to access real-time information, leverage external reasoning systems, and perform meaningful actions in dynamic environments.

Recently, open-sourced research has focused increasingly on enhancing the efficiency of LLMs in tool-use contexts (Qin et al., 2023; Chen et al., 2023; Liu et al., 2024a;b; Zhang et al., 2024a), while also exploring various prompting and training strategies to improve their performance in agentic tasks. Prominent prompting techniques like Chain of Thought (CoT) (Wei et al., 2022), Reflection (Shinn et al., 2024), and ReACT (Yao et al., 2023) have garnered attention. While initial efforts centered on In-Context Learning (ICL)—where pre-trained LLMs were prompted with API specifications and tool-use examples—current approaches are increasingly incorporating fine-tuning methods to enhance model accuracy.

Moreover, popular agent environments such as Webshop (Yao et al., 2022), AgentBench (Liu et al., 2023), WebArena (Zhou et al., 2023), OSworld (Xie et al., 2024), AgentBoard (Ma et al., 2024), BFCL (Yan et al., 2024), and $\tau$-bench (Yao et al., 2024) facilitate agent interactions and evaluations within various scenarios such as web navigation, shopping, games, and computer environments. Specifically, AgentBoard designs two multi-turn environments for tool query and operations, but they only contain 100 user queries in total and do not include real-time feedback. $\tau$-bench simulates dynamic conversations between a user and a language agent, providing API tools and policy guidelines, but it only includes two domains: retail and airline. The LAM Simulator, in particular, aims to create a realistic environment that supports large-scale and diverse tool usages, as well as real-time and multi-turn interactions, providing comprehensive feedback to enhance agent behavior. Table 0 highlights our comparison with popular related frameworks for AI Agents development.

| Framework | Multi-turn | Open Action | Programmatic Evals | Automated Data Gen | Self-exploration |
|---|---|---|---|---|---|
| ToolBench (Qin et al. (2023)) | ✔ | ✔ | ✗ | ✔ | ✗ |
| ToolTalk (Farn & Shin (2023)) | ✔ | ✗ | ✔ | ✗ | ✗ |
| WebArena (Zhou et al. (2023)) | ✔ | ✗ | ✗ | ✗ | ✗ |
| APIGen (Liu et al. (2024b)) | ✗ | ✔ | ✗ | ✔ | ✗ |
| **LAM Simulator (ours)** | ✔ | ✔ | ✔ | ✔ | ✔ |

Table 0: Comparison of Prior Frameworks and Our LAM Simulator. **Multi-turn** assesses support for multi-turn settings, **Open Action** assesses if agent's actions space are predefined or open, **Programmatic Evals** assesses if ALL evaluators (both action and task) are using a programmatic approach without using LLMs, **Automated Data Gen** assesses automated training data generation capabilities, and **Self-exploration** assesses if models can self-improve through the framework without external stronger models or human supervision.

# 3 LAM Simulator Framework

This section outlines the technical details and design principles of the LAM Simulator. The framework is crafted to enable agent models to autonomously explore and enhance their problem-solving abilities. It offers a variety of tasks, access to an extensive set of tools, interactive elements, and automated, high-quality feedback throughout the exploration process. The detail examples of core components are included in Appendix A.2.

## 3.1 Framework overview for Data Generation through online exploration and high-quality feedback

In the LAM Simulator framework, an User Command is a natural language query given by a user to instruct the Agent. The Agent can use multiple tools in sequence to solve tasks. During this process, the Agent assesses the current state to make the decision for next action, makes tool calls, and observes the environment's feedback. This cycle continues until the Agent solves the task or reaches the step limit.

**Generating User Command**: We begin the data generation process by creating User Command data using a Template Matching technique, allowing us to extract key components from the user's command for task evaluation.

First, the Task Manager receives "Content Dataset" that contains data entries as shown at the leftmost block in Figure 1. Each data entry, referred to as "Content data", includes three critical components:

- `task`: The name of the Abstract Task for template matching.
- `user_command_parameters`: Parameters used to generate the User Command with available templates inside Abstract Task.
- `task_available_tools`: Optional tools available for the task. If this component is left blank, the default tools created inside the Abstract Task will be used.

Subsequently, the Task Manager extracts the `task` from the "Content data" to identify the pertinent Abstract Task. Within the designated Abstract Task, the Task Manager searches for the "User

Command Template" that precisely matches the given `user_command_parameters`. For instance, if the "Content data" includes `{"num_apps":5}` and `{"domain":"gaming"}` as its `user_command_parameters`, the Task Manager will locate an "User Command Template" within the Abstract Task that includes these parameters and no others. Assuming the template found is "Show me `num_apps` apps from the app store in `domain`," the Task Manager will generate the User Command "Show me 5 apps from the app store in gaming" to send to the Agent. An error is triggered if no matching User Command Template is found, and consequently, no User Command is created.

If `task_available_tools` are provided, the Task Manager uses this specific set; otherwise, it defaults to the tools in the Abstract Task. The Task Manager queries the Tools Database to extract metadata such as descriptions and parameters. With this information, it generates a User Command that includes the specified tools and their metadata.

**Converting Data and Prepare for generation**: The Task Manager sends the User Command, Task Available Tools, and Final Task Evaluators to create a "Conversation Data" object, which includes conversation history, available tools, and evaluators. The "Conversation Data" is then being sent to the Agent LLM for generating actions.

**Generating data with only interaction and feedback**: Once the "Conversation Data" reaches the Agent LLM, the interaction process begins. The Agent LLM utilizes this data to generate actions and interact with the environment by repeating the following steps until the task is completed or the maximum number of steps is reached:

- Action Generation and Interaction: The Agent LLM produces actions based on its understanding of the task, the available tools, and the current context from the conversation history. It formulates a plan for the current step, specifying tool calls and the corresponding parameters. If the Agent determines it has enough information to provide the final answer to the User Command, it is expected to use a special tool, `Finish`, to wrap up the answer.

- Agent's Action Syntax verification: The Syntax Verification Engine checks that the action commands generated by the Agent are syntactically correct. If they pass the syntax check, the actions are forwarded to the Request Execution Engine. If they fail, feedback is sent back to the Agent for correction. Feedback mechanisms will be discussed in the Evaluators component under 3.2.1.

- Request Execution: The Request Execution Engine, consisting of multiple environments (Env 1, Env 2, etc.), executes the actions given by the Agent LLM.

- Evaluation: The Evaluation Engine evaluates the Agent's responses. It assesses the Agent's action and, if the Agent is at the final step, it evaluates the overall success of the task. Section 3.2.1 will give futher explanation.

- Feedback loop: The feedback from the Evaluation Engine is integrated into the Conversation Data. Additionally, the newly generated actions from the Agent are appended to the conversation history for the next steps in the interaction process.

In summary, the LAM Simulator framework is designed to generate data with high-quality feedback for training agents. It structures interactions using template-based user commands, employs a comprehensive task manager, and offers detailed automated feedback mechanisms. This ensures that the Agent receives continuous and constructive feedback and is thus able to perform its exploration progress to improve agent-task-solving capability without any human intervention. This makes the LAM Simulator a powerful and efficient framework for developing and refining Agentic LLMs.

## 3.2 EVALUATORS

The core functionalities of the LAM Simulator focus on generating timely feedback for agent actions. This subsection discusses the evaluators currently supported by the LAM Simulator.

### 3.2.1 INTERMEDIATE ACTION EVALUATOR

The Intermediate Action Evaluator assesses each action generated by the Agent LLM during intermediate steps of resolving the user request. It takes the Agent's generated text (as a string) and evaluates it through the following steps:

1 Verify that the string is correctly parsed and contains the required components: (1) thought, (2) tool calls, and (3) tool arguments. If this passes, proceed to the next step. Otherwise, a Structure Error is raised.

2 Ensure that the tool call is valid by checking if the tool name is included in the provided list of tools. If this passes, proceed to the next step. Otherwise, a Tool Name error is raised.

3 Validate that the tool arguments are correct for the selected tool calls. This includes ensuring all required arguments are present, no unknown arguments are included, and all arguments are of the correct type. If this passes, the action is marked as successful. Otherwise, a Tool Arguments Error is raised.

The Intermediate Action Evaluator is instrumental in ensuring the Agent operates within the correct parameters throughout its task-solving process. By breaking down the evaluation into distinct checks, LAM Simulator is able to provide precise and actionable feedback at each step.

### 3.2.2 FINAL TASK EVALUATOR

Another key component of the LAM Simulator is the Final Task Evaluator. Our framework automates the evaluation of the Agent's final response to the User Command, eliminating the need for human intervention or powerful Large Language Models. This ensures consistency, efficiency, and scalability.

Each of our Final Task Evaluators takes the `user_command_parameters` and an Agent final response in string format. From here, the Final Task Evaluation is performed as follows:

1 Retrieve Gold Label: The Final Task Evaluator has an internal solution trajectory for each Abstract Task that is hidden from the Agent. Using the `user_command_parameters` as the initial input, the evaluator executes each tool in this solution trajectory sequentially to gather the necessary information to solve the task, known as the `gold label`. In the rare event that any execution fails on the Final Task Evaluator's side, a special message is returned indicating that this evaluation run is invalid.

2 Comparison with Gold Label: The evaluator compares the `gold label` with the Agent's final response. Each task has a unique method for this comparison, which can include exact match, structural match, or key information inclusion. If the Agent's response matches the `gold label` using the defined method, the Task response is marked as "Passed". Otherwise, it is marked as "Failed".

The Final Task Evaluator provides objective and systematic evaluation of the Agent's performance. Utilizing an internal solution trajectory that leverages `user_command_parameters` to generate `gold labels` and predefined comparison methods, the LAM Simulator delivers consistent and fair assessments of the Agent's final responses. This automation reduces dependency on manual evaluations, thereby streamlining the training process.

### 3.3 TOOLS COLLECTIONS

Our toolset comprises a carefully curated collection of 166 high-quality tools that our team manually crafted. These tools come with detailed specifications, including parameter requirements, response formats, and endpoint information. They are designed to integrate seamlessly with agentic LLMs.

In addition to our handcrafted tools, we sourced, filtered, and cleaned-up a significant portion of our tools from ToolBench, resulting in a collection of 3,420 extracted tools.

Details about our procedures of constructing the tools collection are discussed in Appendix A.3.

### 3.4 ABSTRACT TASK CONSTRUCTION

**Human crafted Abstract Tasks**: For the initial version, we support 30 human-crafted tasks. These tasks are meticulously designed by our team to cover common requests an Agent might encounter. (see Figure 2a for detailed statistics).

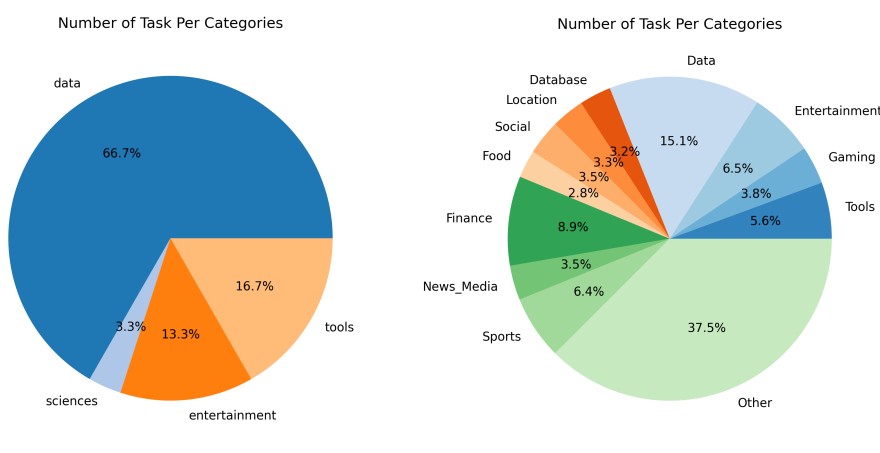

(a) Human crafted Abstract Tasks       (b) ToolBench extracted Abstract Tasks

Figure 2: Distributions for Human crafted Abstract Tasks and ToolBench extracted Abstract Tasks

For each task, we developed a comprehensive set of user command templates that cover a wide range of possible commands an Agent might receive. We also designed thorough evaluations for the Agent's final response for each abstract task. Consequently, all our human-crafted tasks feature both Intermediate Action Evaluator and Final Task Evaluator support.

**ToolBench extracted Abstract Tasks**: To ensure a diverse set of tasks during agent exploration, we incorporated tasks extracted from the ToolBench training dataset. Here, we selected a subset of ToolBench training data that requires tools within our tool collection, resulting in a set of 10,000 tasks, with the distribution illustrated at Figure 2b. These tasks expand the range of tasks available to the Agent and are ideal for probing the Agent's ability to use a diverse collection of tools and identify potential misbehaviors.

While we have collected many of ToolBench extracted tasks, those tasks generated by a Large Language Model (LLM) are much noisier and lower in quality compared to our human-crafted tasks. Therefore, they are only suitable for intermediate steps and are primarily used for exploring tool usage. As a result, only the Intermediate Action Evaluator is applied to these tasks.

At the end of Section 4.3, we will analyze the effectiveness of high-quality data in multi-turn agentic tasks. We will break down the effectiveness into Intermediate steps and Final response steps when solving these tasks. This will underscore the critical role of both high-quality human-crafted tasks and toolbench-extracted tasks in achieving successful outcomes.

In conclusion, by integrating a set of meticulously crafted human tasks with a diverse range of LLM-generated tasks from ToolBench, the LAM Simulator creates a robust training environment. This dual approach ensures that agents are well-equipped to handle a wide array of commands and scenarios, meeting our objective of continuous improvement and automation in LAM training.

## 4 APPLICATIONS

The LAM Simulator excels in automating data generation and delivering real-time feedback on Agent actions, demonstrating its versatility across different scenarios. This section highlights two primary use cases: 1) Generating Preference Data Pairs for DPO Training, and 2) Generating and Filtering High-quality Data for Instruction Finetuning.

### 4.1 UNIFIED EXPERIMENT SETUP

This section outlines the shared experimental procedures and configurations for generating preference data pairs for DPO training and filtering data for instruction fine-tuning as well as the evaluation details of the two following subsections.

**Content Dataset for exploration** We constructed an Content Dataset comprising 400 "Content Data" entries that can be used to generate User command via Template matching using our framework. This dataset includes 166 entries from 30 human-crafted tasks that support both Intermediate Action and Final Task Evaluators, and 234 entries with Intermediate Action Evaluators alone.

**ToolEval**: To evaluate models' performance, we employed the ToolEval benchmark that specifically designed for real-time evaluation of multi-turn reasoning tasks Qin et al. (2023). Here, we conducted evaluations across three distinct scenarios in ToolEval: Unseen Instruction & Seen Tools, Unseen Tools & Seen Categories, and Unseen Tools & Unseen Categories.

**ToolEval-Cleaned** Noting that the ToolEval datasets contains many non-functional tools, we filtered out these tasks for cleaner evaluation. The dataset construction details are discussed in the Appendix.

**Evaluation Metrics**: Following the methodology of ToolEval, we used pass rate as the metric, where the final agent responses are evaluated by GPT-4-0125-preview to determine task successes.

### 4.2 Generating Preference Data Pairs for DPO Training

One notable application is utilizing the action rewards to generate preference data for DPO training.

#### 4.2.1 Experiment Setup

**Base models** We chose two models xLAM-7B-r and xLAM-8x7B-r from the xLAM series (Zhang et al. (2024b)) as our baselines. Despite their strengths in tool usage and function calls, these models sometimes fail. We aimed to use the LAM Simulator to help these models identify and rectify their weaknesses through DPO.

**Generating preference data for DPO** For single-turn use cases, it is straightforward that we can use the same context to generate several different variances of responses and use those variances to create pair-wise data. However, multi-turn conversations present a significant challenge. At an intermediate step $S_i$, an agent may generate several action variations $A_{i1}, A_{i2}, \ldots, A_{in}$, making a complete traversal of all paths computationally infeasible due to exponential growth.

To address this, we developed a Multi-Turn preference data generation method. At each conversation step, instead of always choosing the highest-reward response, we randomly select one response to add to the conversation history. This prevents the agent from encountering only high-reward scenarios, exposing it to a mix of easy and difficult situations. Consequently, the agent learns to manage diverse challenges more effectively. This chosen action then guides the next steps, simplifying the process and reducing computational costs. This method helps agents improve their multi-turn interaction performance by learning from diverse feedback.

**Training data construction** Using 4 H100 GPUs for 8 hours and following our methodologies to generate multi-turn preference data mentioned above, the candidate base models can successfully generate between 300 to 500 pair-wise data points per run.

**Evaluation Metrics**: Following the methodology of ToolEval, we used pass rate as the metric, where the final agent responses are evaluated by GPT-4-0125-preview to determine task successes.

#### 4.2.2 Main Results and Discussions

The results illustrated in the Table 1 demonstrate the significant enhancements achieved by high-performance Large Action Models boosted with the LAM Simulator framework.

For the ToolEval dataset, our LAM-Sim-8x7B model achieved a 14.24% relative improvement over the base model xLAM-8x7B-r and outperformed GPT-4-0125-preview, topping the rankings. LAM-Sim-7B model also showed great improvement, nearing the larger xLAM-8x7B-r's performance.

In the ToolEval-Cleaned dataset, which excludes tasks with non-working tools, the LAM-Sim-8x7B model reached an average pass rate of 52.05%, reflecting an 18.54% relative improvement over the base model. The LAM-Sim-7B model also slightly surpassing the xLAM-8x7B-r's 43.91%, demonstrating our approach's effectiveness in clean testing conditions.

More notably, our LAM-Sim-8x7B achieved a significant performance boost on out-of-domain test scenarios. This highlights its capability in handling varied and complex agentic tasks, demonstrat-

| Model Name | Rank | Avg. ToolEval | U. Inst | U. Tools & U. Cat | U. Tools |
|---|---|---|---|---|---|
| LAM-Sim-8x7B | 1 | **49.86%** | 44.26% | **52.94%** | **52.38%** |
| GPT4-0125-preview | 2 | 46.36% | 46.99% | 52.41% | 39.68% |
| GPT-4o | 3 | 45.80% | 45.36% | 49.20% | 42.86% |
| xLAM-8x7B-r | 4 | 43.65% | 41.53% | 49.20% | 40.21% |
| LAM-Sim-7B | 5 | 41.33% | 39.89% | 45.99% | 38.10% |
| xLAM-7B-r | 6 | 39.33% | 35.52% | 42.78% | 39.68% |
| GPT3.5-Turbo-0125 | 7 | 34.06% | 33.88% | 34.43% | 33.86% |

Table 1: Pass Rate comparison on three distinct categories of ToolEval. Ranking is based on the Average Pass Rate over three scenarios. **Bold** and Underline results denotes the best and second best results for each setting, respectively.

| Model Name | Rank | Avg. ToolEval-Cleaned | U. Inst | U. Tools & U. Cat | U. Tools |
|---|---|---|---|---|---|
| LAM-Sim-8x7B | 1 | **52.05%** | 48.87% | **52.03%** | **55.24%** |
| GPT4-0125-preview | 2 | 47.68% | 51.13% | 51.35% | 40.56% |
| GPT-4o | 3 | 46.22% | 46.62% | 47.97% | 44.06% |
| LAM-Sim-7B | 4 | 43.95% | 46.62% | 42.57% | 42.66% |
| xLAM-8x7B-r | 5 | 43.91% | 45.86% | 44.59% | 41.26% |
| xLAM-7B-r | 6 | 39.81% | 38.35% | 41.22% | 39.86% |
| GPT3.5-Turbo-0125 | 7 | 36.14% | 38.35% | 35.81% | 34.27% |

Table 2: Pass Rate comparison on three distinct categories of ToolEval-Cleaned. Ranking is based on the Average Pass Rate over three scenarios. **Bold** and Underline results denotes the best and second best results for each setting, respectively.

ing its robustness and adaptability in unfamiliar environments. Such versatility in out-of-domain contexts underscores the framework's potential for broader applications and long-term deployment in dynamic and unpredictable settings.

These results underscore the LAM Simulator framework's substantial impact in delivering high-quality data and feedback. This enables Large Action Models to effectively identify and correct their errors, greatly enhancing their agentic capabilities.

### 4.2.3 ERRORS BREAKDOWN ANALYSIS

To understand deeper about how the improvement has been achieved by the high-quality data generated using LAM Simulator, we investigated the errors reduction that we achieved by using LAM Simulator data and feedback.

| Model Name | Avg. Errors | Structure errors | Tool name errors | Tool arguments errors |
|---|---|---|---|---|
| xLAM-7B-r | 11.67 | 14.00 | 2.67 | 18.33 |
| LAM-Sim-7B | 5.11 | 5.67 | 5.67 | 4.00 |
| xLAM-8x7B-r | 8.44 | 10.00 | 2.00 | 13.33 |
| LAM-Sim-8x7B | 5.56 | 8.33 | 0.33 | 8.00 |

Table 3: Error Analysis across all three ToolEval sets. The values indicates the number of average errors (count) generated by each model across three scenarios: U. Inst, U. Tools & U. Cat, U. Tools. The "Base Model" column indicates the original model used for training the new models.

Here, we analyzed the inference results on ToolEval of our trained models and their base models, xLAM model series, to understand their current's behavior when using tools. This analysis highlights the remarkable efficacy of the LAM Simulator in reducing overall agent performance errors. The LAM-Sim-7B model recorded an average error count of 5.11, which is a significant 56.19% reduction compared to the xLAM-7B-r's 11.67 average errors. Similarly, the LAM-Sim-8x7B model achieved a 34.21% reduction in errors, lowering the average to 5.56 from the base model's 8.44.

Across different error categories—Structure Errors, Tool Name Errors, and Tool Arguments Errors—our models particularly excelled in reducing Tool Arguments Errors. This indicates that our LAM models demonstrate improved capability in using tools correctly, underscoring the impact of our framework. These comprehensive improvements highlight the pivotal role of the LAM Simulator in enhancing the precision and reliability of agentic models.

### 4.3 GENERATING AND FILTERING HIGH-QUALITY DATA FOR INSTRUCTION FINETUNING

Another application of the LAM Simulator is its ability to utilize the feedback system for real-time data generation and interaction, producing a high-quality dataset for Instruction Finetuning. This approach enhances LLMs with agentic functionalities without the need for external resources such as human labeling or more powerful models (e.g., GPT-4) for supervised data annotations.

#### 4.3.1 EXPERIMENT SETUP

**Base model** We chose Mixtral-8x7B-Instruct-v0.1 (Jiang et al., 2024) as generic LLM since the model has very low but some capability of performing agentic tasks. We were considering Mistral-7B-Instruct-v0.2 (Jiang et al., 2023) as a candidate as well, however, due to limited resources, we discarded this model as it struggles to generate any valid structured data.

**SFT dataset construction** Using the "Content Dataset" mentioned in Section 4.1, we let the core LLM, which in this case is Mixtral-8x7B-Instruct-v0.1, solving the User commands generated by LAM Simulator by interacting with the LAM Simulator's Environment and collect all intermediate steps and final steps data points that passed the Intermediate Action Evaluators and Final Task Evaluators, respectively. We performed the exploration process with 4 H100s for 12 hours and successfully collected around 1000 data points for SFT process.

#### 4.3.2 MAIN RESULTS AND DISCUSSIONS

| Model Name | Avg. ToolEval | U. Inst | U. Tools & U. Cat | U. Tools |
|---|---|---|---|---|
| LAM-Sim-8x7B-Mixtral | 30.94% | 27.87% | 38.50% | 26.46% |
| Mixtral-8x7B-Instruct-v0.1 | 11.79% | 8.74% | 16.04% | 10.58% |

Table 4: Pass Rate comparison on three distinct categories of ToolEval. LAM-Sim-8x7B-Mixtral is our model trained from Mixtral-8x7B-Instruct-v0.1

| Model Name | Avg. ToolEval-Cleaned | U. Inst | U. Tools & U. Cat | U. Tools |
|---|---|---|---|---|
| LAM-Sim-8x7B-Mixtral | 32.80% | 27.82% | 41.22% | 29.37% |
| Mixtral-8x7B-Instruct-v0.1 | 12.15% | 9.77% | 16.89% | 9.79% |

Table 5: Pass Rate comparison on three distinct categories of ToolEval-Cleaned. LAM-Sim-8x7B-Mixtral is our model trained from Mixtral-8x7B-Instruct-v0.1

The results presented in the table Table 4 and Table 5 reveal significant improvements in performance for the models enhanced with the LAM Simulator framework. On both ToolEval and ToolEval-Cleaned, our LAM-Sim-8x7B-Mixtral model showed an exceptional performance, with pass rate consistently doubled or even tripled the base model, Mixtral-8x7B-Instruct-v0.1, demonstrating its solid capability in solving agentic tasks. This giant improvement without using external supervised data highlights the critical role of high-quality feedback and enriched datasets in advancing LLM performance, especially in complex agentic tasks.

#### 4.3.3 IMPACT OF DIFFERENT REWARD COMPONENTS

We conducted an ablation study to evaluate the impact of different evaluators in our LAM Simulator. Specifically, we analyzed the contributions of our Intermediate Action Evaluator, which measures the quality of the LLM in generating correct actions involving tool usage, and our Final Task Evaluator, which assesses the quality of the LLM's final step in responding to user requests.

To carry out this study, we started with a high-quality dataset (HQ-Data) used to train the LAM-Sim-8x7B-Mixtral, which is based on Mixtral-8x7b-Instruct-v0.1.

We curated a Low-Quality Intermediate step data (LQ-Interim) by adding on top of HQ-Data with actions that were rejected by the Intermediate Action Evaluator. This dataset represents scenarios where tool usage data are not well curated, a function handled by our Intermediate Action Evaluator.

Similarly, we curated Low-Quality Final response scenarios by including final step data rejected by our Final Task Evaluator with our existing HQ-Data, creating the Low-Quality Final Response (LQ-Final) dataset. This dataset represents scenarios where the quality of final responses is not guaranteed, a task usually managed by our Final Task Evaluator.

We then followed the same procedure to develop LAM-Sim-8x7B-Mixtral by running Instruction Finetuning on these two noisy dataset and evaluating the performance across three ToolEval test sets. The average pass rate over all three scenarios was measured and reported.

The results, displayed in Figure 3, show that training the model on the LQ-Interim dataset significantly degrades its performance, essentially nullifying its capability to perform any agentic tasks. This underscores the importance of high-quality action data for tool usage. Additionally, the inclusion of low-quality final responses results in a substantial performance drop, with around a 30% decline in relative performance.

In summary, this ablation study demonstrates that both intermediate and final rewards are essential for the LAM Simulator's ability to develop superior agent models. Intermediate rewards are particularly crucial for enabling generic LLMs to generate valuable training data. Continuous feedback throughout the learning process helps the agent navigate and refine its actions, ultimately leading to improved overall task performance.

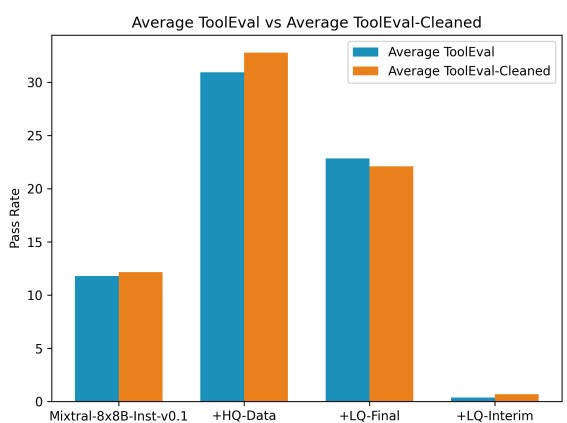

Figure 3: Pass Rate on ToolEval and ToolEval-Cleaned with different data settings: "+HQ-Data" indicates the use of high-quality datasets evaluated by our Evaluators, "+LQ-Final" adds noisy Final Step data to "HQ-Data," and "+LQ-Interim" adds noisy Intermediate Step data to "HQ-Data."

## 5 CONCLUSION

In this paper, we presented LAM Simulator, a comprehensive framework designed to advance the development of Large Action Models (LAMs) by enabling self-learning through online exploration and automated feedback. Our system effectively addresses the limitations of traditional supervised learning and manual data curation, offering a scalable solution that enhances both agentic performance and training efficiency. LAM Simulator provides real-time interactions, multi-turn task processing, and high-quality feedback, contributing to significant improvements in model training performance across various benchmarks, such as ToolEval, where models trained via our simulator outperformed other leading models. Our framework accelerates the learning and adaptation process of LAMs with minimal human intervention, demonstrating its potential as a pivotal tool for future research and development in AI agents.

However, the LAM Simulator still has some limitations. The current implementation focuses on predefined tasks and tools, which may limit its adaptability in more dynamic or unstructured environments. In future work, we aim to expand the framework's generalization capabilities by incorporating a wider range of tasks and tool integrations, as well as exploring methods for better handling ambiguous or incomplete task specifications. Furthermore, we plan to investigate the scalability of the system in environments with more complex action spaces and interdependencies, pushing the boundaries of autonomous agent learning.

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

# A  APPENDIX

## A.1  CONSTRUCTING TOOLEVAL-CLEANED

To ensure the quality of our dataset, we implemented a filtering process to exclude data points that involve one or more non-working tools. This section outlines the high-level procedures for this data filtering method.

We defined a set of keywords associated with non-working tools, such as "API doesn't exist" and other related error messages. These keywords help identify data points with errors in the intermediate steps of the models. This set of keywords is being constructed by manually analyzing the tools usage logs of the tools collection in the evaluation datasets.

Then, we performed traversing each test set and For each test set, we identify data points containing any of the predefined error keywords, which allows us to filter out the data points effectively.

By implementing this filtering procedure, we maintain a high standard of data quality, removing instances with non-functional tools that could otherwise skew the evaluation results of our models. This step is crucial in ensuring the robustness and reliability of our LAM Simulator framework.

## A.2 EXAMPLES

### A.2.1 EXAMPLE OF AN "ABSTRACT TASK" IN LAM SIMULATOR

An Abstract Task stores the relevant information to the task at high-level. It can be used as a "blue-print" to generate User Command as well as providing evaluators & gold label answer to a particular task. Each abstract class includes:

- `task`: The name of the Abstract Task

- `description`: The description of the Task

- `user_command_templates`: All possible User Command Templates for the given tasks ("parameters" are wrapped by the brackets).

- `user_command_parameters`: All possible User Command Parameters can be used inside each data entry of the **Content Dataset**. These are the parameters to be plugged in the User Command Templates.

- `final_answer_format_instruction`: The optional instruction explaining how to structure the **agent's final response**

- `related_apis`: Default tools to use in case there is nothing specified inside the entries of **Content Dataset**

- `solutions`: All possible solution paths to solve this task. This involves the tool names and arguments to be used. Arguments values can be either specified as hard-coded values or null. If specified as null, it will look into the execution states to find the corresponding values.

- `evaluators`: All possible evaluators for this task. This includes what to be used as **Intermediate Action Evaluators** and **Task Final Evaluators**.

```
{
    "task": "get_movie_details",
    "description": "Search a movie detail based on name and return
        the detail information of the movie",
    "user_command_templates": [
        "I've been looking up {movie_detail} about the movie {
            movie_name}. Fun fact: the set of {movie_name} was
            built inside a massive warehouse to create a surreal
            atmosphere!",
        "Provide me the details about {movie_name} movie.",
        ...      # more templates here
    ],
    "user_command_parameters": {
        "movie_name": {
            "type": str,
            "description": "The name of the movie to search for.",
        },
        "movie_detail":  {
            "type": str,
            "description": "The detail of the movie to search for.
                ",
        }
    },
    "final_answer_format_instruction": "In your final answer,
        provide the answer in a dictionary format {"movie_detail":
        ..., "title": ....}",
    "related_apis": ["get_search_movie_for_movie_tools", "
        get_movie_details_for_movie_tools"]
    "solutions": [
        {
            "tool_call": "get_search_movie_for_movie_tools",
```

```
            "arguments": {"movie_name": null}
        }, {
            "tool_call": "get_movie_details_for_movie_tools",
            "arguments": {"id": null}
        }
    ],
    "evaluators": {
        "intermediate_action_evaluators": ["
            standard_syntax_evaluator"],
        "final_task_evaluators": ["standard_multi_evaluator"]
    }
}
```

### A.2.2 EXAMPLE OF DOCUMENTS FOR OUR TOOLS COLLECTION

Here, we want to share an example of a tool `get_search_movie_for_movie_tools` to highlight how we manage the documentation for each of the tools in our Tools collection:

```
{
    "name": "get_search_movie_for_movie_tools",
    "category": "entertainment",
    "description": "This is the subfunction for tool \"movie_tools
        \", you can use this tool to search a given movie id and
        basic information of a given movie name",
    "execution_framework": "lam_simulator",
    "required_parameters": [{
        "name": "movie_name",
        "type": 'STRING',
        "description": "The name of the movie to search for.",
        "default": "",
    }],
    "optional_parameters": []
}
```

For each entry in our tools documentation, we have the following fields:

- `name`: The name of the tool

- `category`: The category of the tool

- `description`: The description of the tool

- `execution_framework`: The environment to execute the tool. It can be `lam_simulator` (our native environment) or any other environment

- `required_parameters`: The list of required parameters of the tool

- `optional_parameters`: The list of optional parameters can be used within the tool

### A.2.3 EXAMPLE FOR A "CONTENT DATASET"

A "Content Dataset" is used to stored parameters data entries. Each data entries contains:

- `task`: The name of the targeted Abstract Task

- `user_command_parameters`: The parameters we need to feed into a valid template inside Abstract Task to generate User command. Note that, here, a valid template means one of the templates in the targeted Abstract Task that contains **ALL** user command parameters stated in this data entry.

- `task_available_tools`: The available APIs we want to use for this data entry. If this is left as empty list, then Abstract Task will use the default APIs inside its `related_apis`

```
[
    {
        "task": "get_movie_details",
        "user_command_parameters": {
            "movie_name": "The Dark Knight",
            "movie_detail": "genres"
        },
        "task_available_tools": [
            "get_search_movie_for_movie_tools",
            "get_movie_details_for_movie_tools",
        ]
    }
]
```

#### A.2.4 EXAMPLE OF GENERATED "USER COMMAND"

Given the Abstract Task and Content Data entry above, we have an example of a generated "User Command" by applying the parameters `movie_name` and `movie_detail` into the first User Command Template from Abstract Task `get_movie_details`.

> **Example of generated "User Command"**
>
> "I've been looking up genres about the movie The Dark Knight. Fun fact: the set of The Dark Knight was built inside a massive warehouse to create a surreal atmosphere!"

**Note**: Gold label also can be created dynamically by the time the User Command got generated. There, the Task Manager takes the `user_command_parameters` from the Content Data entry and feeds that into a possible predefined (hidden) solution path (inside `solutions`) to acquire the gold label final answer.

**Note 2**: At the same time, the Task Manager shares the `intermediate_action_evaluators` and `final_task_evaluators` to evaluate agent's actions.

#### A.2.5 EXAMPLE OF INPUT TO AGENT LLM

Here, we selected an example where the input containing some previous interactions between agent and environment to illustrate the multi-turn & real-time interaction setup. Note that we followed xLAM 1.0 format (Zhang et al. (2024b)) and note that the responses in the history steps are generated by xLAM-8x7B-r:

**Example of Input to Agent LLM (part 1 / 3)**

```
[BEGIN OF TASK INSTRUCTION]
You are AutoGPT, you can use many tools (functions) to do the
    following task.
First I will give you the task description, and your task
    start.
At each step, you need to give your thought to analyze the
    status now and what to do next, with a function call to
    actually excute your step.
After the call, you will get the call result, and you are now
    in a new state.
Then you will analyze your status now, then decide what to do
    next...
After many (Thought-call) pairs, you finally perform the task
    , then you can give your finial answer.

Remember:
1. MOST IMPORTANT, in your response of the "Finish" step, you
    MUST strictly follow the response format of what to be
    written inside "final_answer". In your final answer,
    provide the answer in a dictionary format {"movie_detail"
    : ..., "title": ....}
2. The state change is irreversible, you can't go back to one
    of the former state, if you want to restart the task,
    say "I give up and restart".
3. All the thought is short, at most in 5 sentences.
4. Your action must be calling one of the given tools (
    functions).
5. Your action input must be in json format, where action
    inputs must be realistic and from the user. Never
    generate any action input by yourself or copy the input
    description. Do not add unrelated parameters if not
    needed. Do not add optional parameters when it is not
    required or when these information is not needed.
6. You can do more then one trys, so if your plan is to
    continusly try some conditions, you can do one of the
    conditions per try.

Task description:
You should use functions to help handle the real time user
    querys. Remember:
1. ALWAYS call "Finish" function at the end of the task. And
    the final answer should contain enough information to
    show to the user. If you can't handle the task, or you
    find that function calls always fail(the function is not
    valid now), use function Finish->give_up_and_restart.
2. Do not use origin tool names, use only subfunctions' names
    .

You have access of the following tools:
1: movie_tools
[END OF TASK INSTRUCTION]
```

864
865
866
867
868
869
870
871
872
873
874
875
876
877
878
879
880
881
882
883
884
885
886
887
888
889
890
891
892
893
894
895
896
897
898
899
900
901
902
903
904
905
906
907
908
909
910
911
912
913
914
915
916
917

**Example of Input to Agent LLM (part 2 / 3)**

```
[BEGIN OF AVAILABLE TOOLS]
[{"name": "get_search_movie_for_movie_tools", "description":
    "This is the subfunction for tool \"movie_tools\", you
    can use this tool to search a given movie id and basic
    information of a given movie name", "parameters": {"
    movie_name": {"type": "string", "description": "The name
    of the movie to search for.", "example_value": "", "
    required": true}}}, {... Hidden for space ... }]
[END OF AVAILABLE TOOLS]

[BEGIN OF FORMAT INSTRUCTION]
Your output should be strictly in the JSON format, which
    specifies a list of function calls. The example format is
     as follows. Please make sure the parameter type is
    correct. If no function call is needed, please make
    tool_calls an empty list '[]'.
```{"thought": "the thought process", "tool_calls": [{"name":
     "api_name1", "arguments": {"argument1": "value1", "
    argument2": "value2"}}]}```
[END OF FORMAT INSTRUCTION]

[BEGIN OF QUERY]
I've been looking up genres about the movie The Dark Knight.
    Fun fact: the set of The Dark Knight was built inside a
    massive warehouse to create a surreal atmosphere!
[END OF QUERY]
```

Example of Input to Agent LLM (part 3 / 3)

```
[BEGIN OF HISTORY STEPS]
[{
    "thought": "The user is looking for details about the
        movie 'The Dark Knight'. I will start by searching
        for the movie to get its ID, and then I can fetch the
         complete details of the movie.\n",
    "tool_calls": [{"name": "get_search_movie_for_movie_tools
        ", "arguments": {"movie_name": "The Dark Knight"}}],
    "next_observation": {"error": "", "response": {"id": 155,
         "overview": "Batman raises the stakes in his war on
        crime. With the help of Lt. Jim Gordon and District
        Attorney Harvey Dent, Batman sets out to dismantle
        the remaining criminal organizations that plague the
        streets. The partnership proves to be effective, but
        they soon find themselves prey to a reign of chaos
        unleashed by a rising criminal mastermind known to
        the terrified citizens of Gotham as the Joker.", "
        title": "The Dark Knight"}},
    "user_input": ""
}, {
    "thought": "Now that I have the ID of 'The Dark Knight',
        I can proceed to get the complete details of the
        movie.",
    "tool_calls": [{"name": "
        get_movie_details_for_movie_tools", "arguments": {"
        is_id": 155, "id": 155}}],
    "next_observation": {"error": "", "response": {"title": "
        The Dark Knight", "budget": 185000000, "genres": [{"
        id": 18, "name": "Drama"}, {"id": 28, "name": "Action
        "}, {"id": 80, "name": "Crime"}, {"id": 53, "name": "
        Thriller"}], "revenue": 1004558444, "vote_average": 8
        .517, "release_date": "2008-07-16"}},
    "user_input": ""
}, {
    "thought": "None", "tool_calls": [{"name": "None", "
        arguments": "None"}],
    "next_observation": "APINameParsingError: Failed to parse
         the name of function (api). Check again to ensure: 1
        ) your function name is a function (api) inside '
        available_apis'; 2) your action_input follows the
        format of that given function. Failure encountered:
        None not in task available APIs.",
    "user_input": ""
}]
[END OF HISTORY STEPS]
```

### A.2.6 EXAMPLE OF AGENT RESPONSE

We also share the Agent response by xLAM-8x7B-r for the above example.

> **Example of Agent Response**
>
> ```
> {"thought": "", "tool_calls": [{"name": "Finish", "arguments"
>     : {"return_type": "give_answer", "final_answer": {"
>     movie_detail": "The Dark Knight", "title": "The Dark
>     Knight", "budget": 185000000, "genres": [{"id": 18, "name
>     ": "Drama"}, {"id": 28, "name": "Action"}, {"id": 80, "
>     name": "Crime"}, {"id": 53, "name": "Thriller"}], "
>     revenue": 1004558444, "vote_average": 8.517, "
>     release_date": "2008-07-16"}}}]}
> ```

### A.2.7 EXAMPLE OF RESPONSE FROM ENVIRONMENT

Finally, we share the example for the response from the Environment for the Agent's action above.

```
{
    "done": True,
    "next_observation": "{\"movie_detail\": \"The Dark Knight\", \
        "title\": \"The Dark Knight\", \"budget\": 185000000, \"
        genres\": [{\"id\": 18, \"name\": \"Drama\"}, {\"id\": 28,
         \"name\": \"Action\"}, {\"id\": 80, \"name\": \"Crime\"},
         {\"id\": 53, \"name\": \"Thriller\"}], \"revenue\": 10045
        58444, \"vote_average\": 8.517, \"release_date\": \"2008-0
        7-16\"}",
    "eval_responses": {
        "intermediate_action_evaluators": {
            "standard_syntax_evaluator": {
                "response_structure": {
                    "next_observation": "{\"passed\": true, \"
                        reason\": \"correct format!\"}",
                    "reward": 1.0
                },
                "action": {
                    "next_observation": "{\"passed\": true, \"
                        reason\": \"valid tool call!\"}",
                    "reward": 1.0
                }
                "action_args": {
                    "next_observation": "{\"passed\": true, \"
                        reason\": \"valid parameters!\"}"
                    "reward": 1.0
                }
            }
        },
        "final_task_evaluator": {   # note that this only triggers
             at final step / reached max interaction turns
            "standard_multi_evaluator": {
                "next_observation": "{\"passed\": true, \"reason\"
                    : \"passed!\"}",
                "reward": 1.0
            }
        }
    }
}
```

### A.3 DETAILS ON CONSTRUCTING THE TASKS & TOOLS COLLECTION

#### A.3.1 HUMAN-CRAFTED TASKS & TOOLS

We started our tasks coverage by brainstorming the common requested can be received by Agents, allowing us to come up with four domains:

- **Data**: Involves tasks such as data retrieval, processing, and analysis.
- **Tools**: Focuses on utilizing tools to perform actions or solve tasks.
- **Entertainment**: Covers tasks related to finding information on entertainment options such as movies, or executing entertainment-related actions, like playing games.
- **Sciences**: Involves specialized topics about sciences.

With the domain being determined, we started composing detail tasks for each domain. An example of an Abstract Task about `search_movie_details` is provided in Appendix A.2.1.

With the list of tasks we can support, we then constructing at least one hidden solution path for each of our crafted tasks. We came up with the necessary tools to solve it. During this phase, we are also collecting tools related to each task to broaden our tools collection. We stored each tool and its execution logic, and register the document about that Tools for the Agents to refer to into our system. An example of a tool documentation is provided in Appendix A.2.2.

#### A.3.2 TOOLBENCH EXTRACTED TASKS & TOOLS

In addition to the tools we crafted ourselves, we also obtained many of our tools from ToolBench. ToolBench offers a great collections of tools, with over 16,000 instances from RapidAPI Hub, a marketplace for developers to share APIs in different domains. However, during our quality assessment, we found several tools from this batch to be of poor quality or completely unusable. To ensure only the best tools were used, we implemented a stringent evaluation process, including test runs and detailed reviews of each tool's performance and documentation. Ultimately, this helped us narrow down the selection to 3,420 reliable tools from ToolBench that met our standards for quality and documentation, making them suitable for our needs.

The documents for each of the tools we extracted from ToolBench are then collected and registered in our system. We used the same tool documentation format in Appendix A.2.2 to ensure unification.

