# OpenReview forum: "LAM Simulator: Advancing Large Action Model Training for Agent via Online Exploration and Feedback Simulation"
_ICLR.cc/2025/Conference — Submitted to ICLR 2025_

### Official Review · Reviewer_8sh2 · 2024-10-23

**Soundness:** 3
**Presentation:** 3
**Contribution:** 2
**Rating:** 6
**Confidence:** 3

**Summary:**

This paper introduces the LAM Simulator, a framework designed to enhance the training and development of Large Action Models (LAMs) for AI agents. The LAM Simulator aims to mitigate the reliance on supervised learning and manual data curation by enabling online exploration of agentic tasks with real-time, high-quality feedback. The framework provides a diverse set of tasks, tools, and feedback mechanisms to help agents learn more effectively. The paper demonstrates significant performance improvements using the LAM Simulator, with models such as LAM-Sim-8x7B showing an 18.54% improvement over the base model on the ToolEval benchmark.

**Strengths:**

1) One of the key strengths is the ability of the LAM Simulator to automate feedback and reduce the need for human intervention. This is crucial for scaling agentic models and training on more extensive datasets without the burden of manual curation.
2) The empirical results show significant performance improvements on the ToolEval benchmark. The LAM-Sim-8x7B model consistently outperforms other state-of-the-art alternatives, showcasing the potential of the proposed framework.
3) The LAM Simulator integrates action verification mechanisms (e.g., syntax verification, tool name validation), which significantly reduce errors related to tool usage, helping agents perform more reliably in tasks involving multiple tools.

**Weaknesses:**

1) The tasks used in the evaluation are relatively constrained and do not reflect the complexity of real-world agent tasks. The paper could have included experiments in more diverse environments, such as dynamic multi-agent systems or open-ended tasks, to demonstrate broader applicability.
2) While the framework’s feedback loop is central to its design, the paper does not sufficiently explore its effectiveness. It would have been useful to include ablation studies or case studies showing how feedback at different stages (e.g., intermediate actions versus final steps) influences learning outcomes.
3) The paper does not adequately address how well the LAM Simulator scales to more complex environments or to tasks that require real-time interactions with a broader set of tools. The reliance on predefined tasks and tools may limit its generalization to open-world or unstructured environments.

**Questions:**

1) How does the LAM Simulator handle situations where the task requirements change dynamically, or where tools malfunction during execution? Would it be able to generalize to such cases?
2) Have you considered applying the LAM Simulator to environments that involve real-time multi-agent interactions or more complex tool dependencies?
3) Can you provide more detailed insights into how the feedback mechanisms evolve as the agent performs tasks over time? Are there diminishing returns to feedback as the agent improves?
4) How does the action verification mechanism scale with an increasing number of tools or tool parameters? Does it introduce any computational overhead that might limit real-time applicability?

---

> ### Author Response · Authors · 2024-11-21
> **Rebuttal by Authors (part 1)**
>
> Thank you very much for providing valuable feedback and posing excellent questions regarding our paper. We have thoughtfully reviewed your concerns and are eager to respond to them effectively.
>
>
> ### **W1. Complexity and diversity of agent tasks in the evaluation set**
>
> Once again, thank you for your feedback. We fully acknowledge the importance of testing frameworks in varied and dynamic settings, as this not only enhances the generalizability of our findings but also better represents the complicated nature of real-world applications.
>
> While we strive to bridge the gap between academic evaluations and real-world applicability, we chose to utilize the ToolBench[1] benchmarking suite for several reasons that align with both the focus of our tools diversity and the supports for multi-turn tasks evaluation:
>
> 1) API Diversity: ToolBench supports thousands of APIs from RapidAPI Hub marketplace, offering a wide array of real-world utilities spanning various domains. This extensive collection allows us to evaluate the framework's performance across a diverse set of tasks, measuring how well it adapts and responds to different challenges posed by these varying contexts.
>
> 2) Complex Task Scenarios: The tasks within ToolBench[2] require multi-step processing and the coordinated use of multiple tools. Such tasks necessitate detailed planning and dynamic adjustment based on former interactions, closely paralleling the complexity encountered in real-world agent operations.
>
> To illustrate the nature of these tasks, consider this example from our benchmark test set:
> ```
> {
>     "query": "My company is hosting a rugby event and we need to provide pre-match form information to the participants. Can you fetch the pre-match form for a specific rugby match? We would also like to see the incidents that occurred during the match."
>
>     "available_tools": ['leaguenextmatches_for_rugbyapi2', 'leaguemedia_for_rugbyapi2', 'categories_for_rugbyapi2', 'categorytournaments_for_rugbyapi2', 'leaguelogoimage_for_rugbyapi2', 'teammedia_for_rugbyapi2', 'matchincidents_for_rugbyapi2', 'match_for_rugbyapi2', 'prematchform_for_rugbyapi2', 'categoryschedules_for_rugbyapi2']
> }
> ```
> ### **W2. Impact of feedback at different stages on learning outcomes**
>
> Thank you for raising the concerns about the effectiveness of feedback at different stages on the learning outcomes. We have conducted an experiment in Section 4.3.3 to gain the insights about this, and we are really happy to share with you some more insights we have:
>
> Starting with the high-quality dataset (HQ-Data) that we generated using the Mixtral-8x7b-Instruct-v0.1 model, we augmented new dataset to understand the contributions of each evaluator as follows:
>
> 1. To understand impact of Intermediate Action Evaluators, we created a Low-Quality Intermediate Dataset (LQ-Interim) by adding on top of our HQ-Data with actions that were rejected by the Intermediate Action Evaluators.
> 2. To understand impact of Final Task Evaluators, we created a Low-Quality Final Response Data (LQ-Final) by adding on top of our HQ-Data with actions rejected by our Final Task Evaluators.
>
> By running instruction finetuning on Mixtral-8x7b-Instruct-v0.1 using the 3 datasets above, we can see the importances of our evaluators.
> From the outcomes, it's evident that both types of feedback—intermediate and final—are vital.
>
> * Action feedbacks through Intermediate Action Evaluators allows agents to adjust behaviors during each action, such as ensuring it follows correct response format or using correct tools and arguments without hallucinating. This contributes to enhance tool usage abilities. Thus, without this correct feedback on Action-level, data generation through self-exploration is impossible as the generated data can carry a lot of different types of uncontrollable errors. In our experiment, model trained with LQ-Interim shows no capability to solve agentic tasks, highlighting the importance of the Action feedbacks.
> * Task feedbacks through Final Task Evaluators on the other hand allows agents to understand whether it solves this task correctly, or its action needed to be revised. Without the correct feedback on Task-level, data generation through self-exploration is hard to control the quality. In our experiment, model trained with LQ-Final has a 30% relative performance decline comparing to the model trained with HQ-Data.
>
> Hence, ensuring high-quality feedback at both the intermediate action levels and the final task completion stages is essential for the optimal training of agents.
>
> ### **References**
> [1] Qin, Y., Liang, S., Ye, Y., Zhu, K., Yan, L., Lu, Y., Lin, Y., Cong, X., Tang, X., Qian, B., Zhao, S., Hong, L., Tian, R., Xie, R., Zhou, J., Gerstein, M., Li, D., Liu, Z., & Sun, M. (2023, July 31). ToolLLM: Facilitating large language models to master 16000+ real-world APIs. arXiv.org. https://arxiv.org/abs/2307.16789

---

> ### Author Response · Authors · 2024-11-21
> **Rebuttal by Authors (part 2)**
>
> Below are the second part of our response.
>
> ### **W3. Reliance on pre-defined tasks & tools / Generalization to open-world or unstructured environment**
> When evaluating the models went through our framework for data generation & training, we are using 3 ToolEval test sets from ToolBench[1], which are:
> 1) U.Inst (Unseen Instruction & Seen Tools): this set contains tasks involving **unseen** instruction (or user command query) with tools **seen** during training phase
> 2) U.Tools (Unseen Tools & Seen Categories): this set contains tasks involving only **unseen** tools from **seen** task category
> 3) U. Tools & U.Cat (Unseen Tools & Unseen Categories): this set contains tasks of **unseen** tools from **unseen** task category
>
> The performance results from these tests have been significantly positive. Notably, our models experienced exceptional improvements in the "U.Tools" and "U.Tools & U.Cat" categories (refer to Tables 1, 2, 4, and 5). This highlights the robust out-of-domain performance and generalization capabilities of our LAM Simulator, evidencing its effectiveness in reducing tool-related errors and enhancing agent reliability across diverse tasks.
>
>
> ### **Q1. How does the LAM Simulator handle situations where the task requirements change dynamically, or where tools malfunction during execution? Would it be able to generalize to such cases?**
>
> The core value of LAM Simulator's design is the concept of promoting the Agent's capability for self-exploration. This approach is fundamental in enabling the Agent to navigate through varying scenarios, rather than merely learning to solve a predefined set of tasks via fixed paths. Here’s how we’ve implemented this:
>
> 1) **Intermediate Action Evaluator**: this evaluator checks for syntactical errors, including response structure, tool calls, tool arguments. If there are extra requirements for each action given by the task, this evaluator also looks at the agent's action and observation from the Request Execution Engine to assign the score.
> 2) **Final Task Evaluator**: this evaluator only triggers when the agent makes the final response to the given multi-turn task or when the number of interaction turns reaches a predefined limit. We compare the final result (state) between the gold label and the final response provided by the agent to give the corresponding score.
>
> By separating the evaluation into these two stages, with the Intermediate Action Evaluator providing necessary checks during each step while solving the task and the Final Task Evaluator assessing the final answer, we allow the Agent the flexibility to explore various approaches to solving the task without being restrainted to a predefined solution path.
>
> Furthermore, the environments integrated into the LAM Simulator are programmed for real-time tool execution, which not only simulates but actively engages the Agent in scenarios where tool malfunctions may occur. This setup is crucial as it necessitates that Agents adapt their strategies based on the current operational state of their tools—essentially training them to anticipate and rectify issues dynamically, fostering resilience and versatility.
>
> This environment thereby supports Agents in developing strategies that are not only effective in familiar settings but are also robust and adaptable in facing new or unforeseen challenges. Through this design, our goal is to enable the Agent to generalize effectively to new tasks and handle unexpected situations in real-time with greater competence.
>
> We hope this explanation addresses your concerns, and we are open to further discussions to enhance our system's responsiveness to dynamic task requirements and tool functionalities. Thank you for your constructive feedback.
>
> ### **References**
> [1] Qin, Y., Liang, S., Ye, Y., Zhu, K., Yan, L., Lu, Y., Lin, Y., Cong, X., Tang, X., Qian, B., Zhao, S., Hong, L., Tian, R., Xie, R., Zhou, J., Gerstein, M., Li, D., Liu, Z., & Sun, M. (2023, July 31). ToolLLM: Facilitating large language models to master 16000+ real-world APIs. arXiv.org. https://arxiv.org/abs/2307.16789

---

> ### Author Response · Authors · 2024-11-21
> **Rebuttal by Authors (part 3)**
>
> Below are the last part of our response.
>
> ### **Q2. Have you considered applying the LAM Simulator to environments that involve real-time multi-agent interactions or more complex tool dependencies?**
>
> **About multi-agent interactions**: In this work, the simulation focuses on single-agent scenarios, due to our prioritization of refining single agent interactions. However, the architecture of the LAM Simulator is designed to be scalable and allows for eventual extension to multi-agent dynamics. This will enable us to support real-time multi-agent interactions in future iterations of the simulator.
>
> **Complex Tool dependencies**: In developing the LAM Simulator, we acknowledged the complexities of tool dependencies across various environments. An engine was implemented to standardize interactions, acting as a mediator to format agent actions and environment observations appropriately. To integrate new environments with complex tool dependencies, we simply need to register the pre-processing and post-processing logic for each specific environment.
>
> We appreciate your feedback as it reinforces the importance of these features and guides the future development of the LAM Simulator. We are excited about the potential to expand into multi-agent settings and further enrich the simulation capabilities.
>
> ### **Q3. Can you provide more detailed insights into how the feedback mechanisms evolve as the agent performs tasks over time? Are there diminishing returns to feedback as the agent improves?**
>
> For any given set of tasks, as the agent's performance improves and it becomes more adept at handling these tasks, the effectiveness of feedback on these set of tasks diminishes. This means that while the agent initially gains significant insights from feedback, over time, its utility decreases as the task complexity remains static. To address this, our LAM Simulator is designed to easily extend to new environments and tasks. This adaptability allows us to introduce sophisticated tasks with corresponding evaluators, enabling agents to continually enhance their capabilities.
>
> As future work, we are planning to support more complicated environments as well, and hoping that we can see other potential directions such as enabling curriculum learning for training agent models, where the tasks difficulty continuously got leverage during training to improve the agent’s quality.
>
> ### **Q4. How does the action verification mechanism scale with an increasing number of tools or tool parameters? Does it introduce any computational overhead that might limit real-time applicability?**
>
> The verification mechanism only verify one tool per step (the tool got called by the agent), so despite the scale of available tools, we will not get blocked by the verifier. This would result in no extra overhead to real-time applicability as we support more tools.
>
> ### **References**
> [1] Qin, Y., Liang, S., Ye, Y., Zhu, K., Yan, L., Lu, Y., Lin, Y., Cong, X., Tang, X., Qian, B., Zhao, S., Hong, L., Tian, R., Xie, R., Zhou, J., Gerstein, M., Li, D., Liu, Z., & Sun, M. (2023, July 31). ToolLLM: Facilitating large language models to master 16000+ real-world APIs. arXiv.org. https://arxiv.org/abs/2307.16789

---

> > ### Author Response · Authors · 2024-12-03
> > **Thank you for your review!**
> >
> > We sincerely thank you for reviewing our paper and increasing the score to 6! Your feedback means great to us, and we remain dedicated to improving our work further.
> >
> > With best regards,
> >
> > Authors of submission 13418

---

### Official Review · Reviewer_nYeX · 2024-11-03

**Soundness:** 3
**Presentation:** 3
**Contribution:** 3
**Rating:** 6
**Confidence:** 4

**Summary:**

The paper introduces the LAM Simulator, a framework for enhancing Large Action Model (LAM) training through online exploration and simulated feedback. By providing AI agents with curated tools, feedback, and interactive environments, the framework enables cost-effective, autonomous data generation with minimal human input. Experiments show significant performance improvements, particularly with the LAM-Sim-8x7B model, and error reduction through automated feedback.

**Strengths:**

1.	**Innovative Framework:** The LAM Simulator introduces a novel method for automating LAM training through simulated feedback and task exploration.
2.	**Performance Improvements:** The framework achieves measurable improvements over baseline models, e.g. GPT-4o, showing its effectiveness in enhancing agent task performance.
3.	Scalability: The reduction in human intervention for data generation makes this framework adaptable for large-scale agent training.
4.	**Comprehensive Evaluation:** The use of multiple evaluators provides continuous feedback, leading to improved agent performance and fewer errors.

**Weaknesses:**

1. **Limited Comparative Analysis:** While the paper briefly mentions related methods like ToolTalk, WebArena, and APIGen, it lacks a thorough comparison with these and other frameworks for automated data synthesis and agent training. Such a comparison would provide valuable context and clarity on the specific advantages or disadvantages of the LAM Simulator relative to existing approaches. The authors could strengthen the paper by providing empirical comparisons or, at minimum, a more detailed discussion of how the LAM Simulator diverges or improves upon these methods.

2. **Comparative Performance Analysis on Agent Self-Exploration:** Although the paper demonstrates improvements over xLAM, it would benefit from a direct, quantitative comparison to other self-exploration-based methods. This could include metrics for data quality, agent adaptability, or overall effectiveness in reducing human intervention. Without such benchmarks, it’s challenging to assess how the LAM Simulator stacks up against state-of-the-art methods in self-guided agent learning.

3. **Insufficient Examples of Human-Crafted Tools:** The paper describes a curated set of tools used for agent training but doesn’t provide concrete examples or details on these tools in the main text or the appendix. Including examples, especially in an appendix, would improve transparency around the toolset design and selection criteria. This addition would allow readers to better understand the diversity and complexity of tasks the agents are trained on, as well as any limitations or biases in the tool curation process.

**Questions:**

1. **Lack of Detail on Reward Calculation:** The paper briefly describes using “action rewards” to generate preference data for DPO but doesn’t clarify how these rewards are calculated. Are rewards based on a specific scoring metric, or are they derived from evaluator feedback? A clearer explanation of the reward components and their calculation method would clarify how preference scores are assigned to different actions or paths.

2. **Potential Overfitting to ToolEval Benchmark:** With performance gains reported on ToolEval, there’s a possibility that the LAM Simulator is optimized specifically for this benchmark. Given ToolEval’s structured and predefined nature, the model might be learning patterns specific to ToolEval, which could limit generalizability. Testing the framework on multiple diverse benchmarks, or introducing a new, unstructured benchmark, would give a more robust picture of its true capabilities.

3. **Limited Discussion on the Impact of Feedback Granularity:** The framework provides both action-wise and task-wise feedback, but the paper does not analyze the relative impact of these feedback types on agent learning. Understanding whether detailed, step-by-step feedback leads to more effective training compared to feedback provided only at task completion could help optimize the feedback strategy and reduce computation costs.

4. **What is the success rate of data generation?:** Can the authors provide quantitative information on the success rate of data generation within the LAM Simulator? For instance, out of all attempted tasks or interactions, what percentage completes successfully without errors? This metric would give insight into the reliability and efficiency of the simulator.

5. **What specific measures ensure the quality of generated data?:** The paper mentions filtering methods to ensure data quality, but could the authors elaborate on the full set of criteria or techniques used to evaluate data quality? Are there specific metrics, benchmarks, or thresholds that generated data must meet?

---

> ### Author Response · Authors · 2024-11-23
> **Rebuttal by Authors (part 1)**
>
> We deeply appreciate your valuable feedback and insightful questions regarding our paper. We have thoughtfully reviewed your concerns and are eager to provide you with responses.
>
> ### **W1. Limited Comparative Analysis**
> Thank you for commenting about the comparison to related work. We acknowledge the importance of a comprehensive comparative analysis and understand that having one would offer a clearer perspective of our framework’s positioning relative to existing technologies.
>
> To address this, we have added the comparison into our revised submission to include a more detailed comparison with other frameworks in Section 2. Here is the attached comparison table for your quick overview.
>
>
> |                          | Multi-turn | Open Action            | Programmatic Evals  | Automated Data Gen       | Self-exploration |
> |--------------------------|------------|------------------------|-----------------------|--------------------------|------------------|
> | ToolBench             |  **✓**     |  **✓**                 | ✗                     |  **✓**                   | ✗                |
> | ToolTalk              |  **✓**     | ✗                      |  **✓**                | ✗                        | ✗                |
> | WebArena              |  **✓**     | ✗                      | ✗                     | ✗                        | ✗                |
> | APIGen                | ✗          |  **✓**                 | ✗                     |  **✓**                   | ✗                |
> | **LAM Simulator (ours)** |  **✓**     |  **✓**                 |  **✓**                |  **✓**                   |  **✓**           |
>
> Here, we display the comparison of Prior Frameworks and Our LAM Simulator. **Multi-turn** assesses support for multi-turn settings, **Open Action** assesses if agent’s actions space are predefined or open, **Programmatic Evals** assesses if ALL evaluators (both action and task) are using a programmatic approach without using LLMs, **Automated Data Gen** assesses automated training data generation capabilities, and **Self-exploration** assesses if models can self-improve through the framework without external models or human supervision.
>
> We believe this comparison not only clarifies the positioning of our framework but also substantiates the LAM Simulator's contributions to the field. We hope this revision will satisfy the need for a clear contextual understanding of our work in relation to existing contributions.
>
> ### **W2. Comparative Performance Analysis on Agent Self-Exploration**
>
> Thank you for your feedback. We recognize your suggestion for a comparative analysis with other self-exploration methods.
>
> To the best of our knowledge, there are rare literature for self-exploration work in the LLM agent space, except some very recent ones. Though, it would be challenging and take siginificant efforts to applying them under the reasonable setting for a fair and completed comparison. Therefore, they are beyond the current scope of this paper and we leave it as an interesting future direction. However, we would like to emphasize that our work is focusing to remove the reliance on human or stronger models to generate high-quality training data for agent development.
>
> With the aid of the simulated framework that enables real-time interactions and our evaluators (both action and task), we are able to offer the capability for LLMs to self-explore and build a high-quality training dataset directly from a base model, which then significantly improves its own performance, as highlighted in Section 4 of our paper.
>
> ### **W3.Insufficient Examples of Human-Crafted Tools**
>
> Once again, thank you so much for bringing up this point. We have addressed this in our Rebuttal revision by adding a details about how we have created the Tools collection in Appendix A.3.
>
> We have also included examples of how we store documentation for each tool in our Tools collection in Appendix A.2.2 to give more detail of how can Agent access tools information.

---

> ### Author Response · Authors · 2024-11-23
> **Rebuttal by Authors (part 2)**
>
> Below is our second part of the response.
>
> ### **Q1. Lack of Detail on Reward Calculation**
> Thank you so much for raising the question about how our rewards are calculated. We are more than happy to provide a more detail explanation regarding to the interaction flow between Agent and our framework and how do we assign reward on the go:
>
> For each of the Agent step to solve a given task, the Agent gives a response (include a Tool call and arguments) to the Environment. This first goes through **Syntax Verification Engine** to check syntactical issues (format, valid tool call, valid args). Here, if anything failed, we share this information to **Evaluation Engine** and the action would have the binary score of 0 (Failed). Otherwise, we can send the Agent’s action to the **Request Execution Engine** to execute the request then sends the corresponding observation to the **Evaluation Engine** for evaluating the executed action.
>
> In Evaluation Engine, we have 2 different evaluators:
>
> 1. **Intermediate Action Evaluator**: this evaluator assigns a binary score (0: Failed, 1: Passed) based on the syntactical feedback and any task-specific requirements.
> 2. **Final Task Evaluator**: this evaluator only triggers when the agent makes the final response to the given multi-turn task or when the number of interaction turns reaches a predefined limit. We compare the final result (state) between the gold label and the final response provided by the agent to give the corresponding binary score (0: Failed, 1: Passed).
>
> In this work, we selected only the generation outputs that passed ALL evaluators for positive training data, and others can be selected for negative training data in case of DPO training.
>
> After syntax checking, executing actions, and scoring, the Agent's action and the Environment observations are sent to the Conversation Data Manager to either proceed to the next turn or finalize the current task and move to the next one.
>
> ### **Q2. Potential Overfitting to ToolEval Benchmark**
> To ensure a fair and unbiased assessment of the LAM Simulator, we have implemented a strict separation in our data handling and task generation processes. Specifically, for the human-crafted tasks, they were developed independently of ToolBench through a separate pipeline that does not interact with or utilize data from ToolEval, thereby preventing any inadvertent exposure to the benchmark during training. Furthermore, for tasks extracted from ToolBench, we strictly avoid using tasks, tools, or domains that are part of the ToolEval benchmark. This rigorous approach guarantees that our training set remains distinct and unbiased, ensuring that the evaluation on ToolEval genuinely reflects the model's ability to generalize.
>
> The results presented in our paper under Section 4 are run on different test sets from ToolEval, including:
> 1. U.Inst (Unseen Instruction & Seen Tools): this set contains tasks involving **unseen** instruction (or user command query) with tools **seen** during training phase
> 2. U.Tools (Unseen Tools & Seen Categories): this set contains tasks involving only **unseen** tools from **seen** task category
> 3. U.Tools & U.Cat (Unseen Tools & Unseen Categories): this set contains tasks of **unseen** tools from **unseen** task category
>
> The results from these tests have been very positive, including out-of-domain testings. In particular, our models showed significant improvements in the "U.Tools" and "U.Tools & U.Cat" categories (see Tables 1, 2, 4, and 5). This demonstrates that our LAM Simulator performs well even in new and different situations, effectively reducing errors related to tools and making the system more reliable for a variety of tasks.
>
> Additionally, we have normalized the input by structuring the Agent's template to align with its base model's architecture, ensuring all tasks from any environment, whether native, from ToolBench, or newly introduced, adhere to a standard format. This template unification allows us to focus on the Agent's capabality in general instead of optimizing towards any given set of tasks or environments.

---

> ### Author Response · Authors · 2024-11-23
> **Rebuttal by Authors (part 3)**
>
> Below is our third part of the response.
>
> ### **Q3. Limited Discussion on the Impact of Feedback Granularity**
> Thank you for raising the concerns about the effectiveness of feedback at different stages on the learning outcomes. We have conducted an experiment in Section 4.3.3 to gain the insights about this, and we are really happy to share with you some more insights we have.
>
> Starting with the high-quality dataset (**HQ-Data**) that we generated using the Mixtral-8x7b-Instruct-v0.1 base model, we augmented new dataset to understand the contributions of each evaluator as follows:
>
> + To understand impact of Intermediate Action Evaluators, we created a Low-Quality Intermediate Dataset (**LQ-Interim**) by adding on top of our HQ-Data with actions that were rejected by the Intermediate Action Evaluators.
>
> + To understand impact of Final Task Evaluators, we created a Low-Quality Final Response Data (**LQ-Final**) by adding on top of our HQ-Data with actions rejected by our Final Task Evaluators.
>
> By running instruction finetuning on Mixtral-8x7b-Instruct-v0.1 using the 3 datasets above, we can see the importances of our evaluators. From the outcomes, it's evident that both types of feedback—intermediate and final—are vital:
>
> + Action feedbacks through Intermediate Action Evaluators allows agents to adjust behaviors during each action, such as ensuring it follows correct response format or using correct tools and arguments without hallucinating. This contributes to enhance tool usage abilities. Thus, without this correct feedback on Action-level, data generation through self-exploration is impossible as the generated data can carry a lot of different types of uncontrollable errors. In our experiment, model trained with LQ-Interim shows no capability to solve agentic tasks, highlighting the importance of the Action feedbacks.
> + Task feedbacks through Final Task Evaluators on the other hand allows agents to understand whether it solves this task correctly, or its action needed to be revised. Without the correct feedback on Task-level, data generation through self-exploration is hard to control the quality. In our experiment, model trained with LQ-Final has a 30% relative performance decline comparing to the model trained with HQ-Data.
>
> Hence, ensuring high-quality feedback at both the intermediate action levels and the final task completion stages is essential for the optimal training of agents.

---

> ### Author Response · Authors · 2024-11-23
> **Rebuttal by Authors (part 4)**
>
> This is our last part of the response.
>
> ### **Q4. What is the success rate of data generation?**
>
> We designed the data generation strategy as a self-learning procedures, in which we let the Agent LLMs to continuously solve given tasks and record the qualified data for training purpose. To show the effectiveness of our system on both top-performing Agent LLMs and low-peroformance LLMs, we are going to display the base Agent's pass rate when running data generation.
>
> **Setup**:
> - We used a **Content Dataset** of 400 entries. This contains 166 entries for our human-crafted tasks and 234 entries for our automated tasks extracted from ToolBench.
> These data entries are ultimately used to fit in the corresponding Tasks' templates to create final User Command. We have examples in Appendix A.2 to illustrate this process.
>
> - For each task, we let the agent to continuously interact with the environment to solve the task, in which for each step, the agents creates 7 generations with sampling temperatures range from 0 to 1. Each generation is then feeded into our evaluators to get a binary score of either 0 for Failed or 1 for Passed.
> We then calculate the number of passed for Actions and Tasks as follows.
>
> **Pass rate illustration**:
> As mentioned above, we are now displaying the results for pass rate on both Action-wise and Task-wise.
>
> **Note** that, only our human-crafted tasks are equipped with Task evaluator, so we are only recording the Task pass rate for tasks under this type.
>
> - **xLAM-7B-r[5] (base model for LAM-Sim-7B) Action and Task pass rate**
>     | Task type     | Action | Task   |
>     |---------------|--------|--------|
>     | Human-crafted | 0.9135 | 0.5783 |
>     | Extracted     | 0.8116 | NaN    |
>
> - **xLAM-8x7B-r[5] (base model for LAM-Sim-8x7B) Action and Task pass rate**
>     | Task type     | Action | Task   |
>     |---------------|--------|--------|
>     | Human-crafted | 0.9130 | 0.6325 |
>     | Extracted     | 0.8374 | NaN    |
>
> - **Mixtral-8x7B-Instruct-v0.1[6] (base model for LAM-Sim-8x7B-Mixtral) Action and Task pass rate**
>     | Task type     | Action | Task   |
>     |---------------|--------|--------|
>     | Human-crafted | 0.3960 | 0.2590 |
>     | Extracted     | 0.3895 | NaN    |
>
> In this setup, we use passed data to create training datasets. Models with lower success rates require more time to produce sufficiently large training datasets, which in return, these trainining sets can improve the models more significant.  The tables above also show that even high-quality Agent LLMs like the xLAM series still have struggle with using tools correctly for each action and completing tasks accurately. This highlights the strength of our framework and its task collection in identifying and fixing errors systematically in a self-exploration way, which is critical to reduce human intervention when developing AI Agents.
>
> ### **Q5. What specific measures ensure the quality of generated data?**
> In this work, we only selected the positive data points for training using only the ones passed all of our evaluators (Intermediate Action Evaluators and Final Tasks Evaluators). Since our evaluator is systematic and got implement to correctly capture the expected state and response state from agent, we would be able to ensure the quality of the response.
>
> Once again, thank you so much for all of your valuable feedbacks. We hope our responses have addressed your concerns and questions well!

---

> > ### Comment · Reviewer_nYeX · 2024-12-02
> >
> > Sorry for the late reply, and thank you for your detailed responses and for addressing my concerns. I appreciate the additional clarifications and the revisions you’ve made to the manuscript. After reviewing the changes, I have decided to maintain my original rating.

---

> > > ### Author Response · Authors · 2024-12-03
> > > **Thank you for your review!**
> > >
> > > Thank you for reviewing our paper and maintaining the score! We greatly appreciate your feedback and remain dedicated to improving our work.
> > >
> > > With best regards,
> > >
> > > Authors of submission 13418

---

### Official Review · Reviewer_4LXX · 2024-11-03

**Soundness:** 2
**Presentation:** 2
**Contribution:** 2
**Rating:** 6
**Confidence:** 2

**Summary:**

The submission introduces a system that incorporates a simulator with a language model for more effective large action models. In response to a user query the LAM Simulator system iteratively refines its response multiple times in a loop. The refinement leverages feedback from the simulator which has syntax verification and request execution tools to improve the final response.

**Strengths:**

- Preference data generation for finetuning via DPO is neat to address the challenges of multi-turn conversations, it encourages more diverse data pairs
- Compared to past work this submission introduces many more domains the system can handle queries for

**Weaknesses:**

- Many parts of the submission seem to lack explanation or additional clarification (perhaps to be put in the appendix) on many details. I ask some direct clarification questions in the questions section. It is also possible that my inexperience with research on these types of agents has led to me being unaware of some terms.
- Figure 1 seems to suggest you can only pass or fail via the syntax verification engine. What happens after the request verification engine?
- There's a lack of example/qualitative user interactions and agent responses (including the generated conversation data in the multi-turn setup). There also appear to be no examples of user command templates, task evaluators etc., (many elements of figure 1). It is quite hard to visualize what exactly is going on and how it might be better than related work.

**Questions:**

- How come the human-crafted abstracted tasks has such a relatively high proportion of "astronomy" tasks. It feels like strangely very specific, how come there aren't other e.g. science topics? It is claimed in section 3.4 the human-crafted abstracted tasks are meticulously designed for the typical requests the LAM might receive from users. However this seems heavily audience dependent, perhaps I missed a sentence somewhere but is the audience mostly astronomers in this case?
- For task categories what does "data" and "database" mean exactly? Why is the "data" category such a high proportion?
- Why DPO for preference optimization instead of other options?
- What is U. Tool, U. Cat etc in the header of table 1?
- Table 2 shows average errors, but these numbers are meaningless without knowing the number of times the model was evaluated. Moreover, there are no standard deviation metrics in any tables and specifically table 2, it is difficult to know if the results are really statistically significant or if there is some lucky prompting changes/randomness in the inference models or evaluation LLMs.

If all weaknesses and questions are addressed I am happy to raise the score into the accept range. My biggest concern is mostly around the lack of examples and explanations for a lot of terms and phrases.

---

> ### Author Response · Authors · 2024-11-21
> **Rebuttal by Authors (part 1)**
>
> We sincerely thank you for sharing a lot of great insights about our paper as well as raising amazing questions for us. We have carefully considered your concerns and really hope we can address those to you.
>
> ### **W1. Figure 1 seems to suggest you can only pass or fail via the syntax verification engine. What happens after the request verification engine?**
>
> Our LAM Simulator framework focuses on providing high-quality feedback for each agent’s action, which can be divided into several layers:
>
> First, the agent’s response (containing a Tool call and its corresponding arguments) is sent through the **Syntax Verification Engine** to check if: 1) the response format is correct, 2) the Tool call is valid (i.e. provided to the agent), and 3) the corresponding Tool arguments are correctly used.
>
> If the agent’s response fails here, the **Syntax Verification Engine** sends this information directly to the **Evaluation Engine** to assign the corresponding score given this failure. Otherwise, we send the agent’s Tool call to the **Request Execution Engine** to execute the request then sends the corresponding observation to the **Evaluation Engine** for evaluating the executed action.
>
> In **Evaluation Engine**, we have 2 different evaluators:
>
> 1. **Intermediate Action Evaluator**: this evaluator assigns the score based on the syntactical feedback from the Syntax Verification Engine mentioned above. If there are extra requirements for each action given by the task, this evaluator also look at the agent's action and observation from the Request Execution Engine to assign the score.
> 2. **Final Task Evaluator**: this evaluator only triggers when the agent makes the final response to the given multi-turn task or when the number of interaction turns reaches a predefined limit. We compare the final result (state) between the gold label and the final response provided by the agent to give the corresponding score.
>
> The scores from Intermediate Action Evaluator and (optionally) Final Task Evaluator are used to determined whether we want to use this generation output from agent as a training data instance. In this work, we selected only the generation outputs that passed ALL evaluators for positive training data, and others can be selected for negative training data in case of DPO training.
>
> After finishes checking the syntax, executing the agent’s action, and assign corresponding scores, the current Agent’s action, observation from environment, and scores from evaluators will finally be sent back to the Conversation Data manager to either proceed with the next turn or finalize this task and move on to the next one.
>
> ### **W2. Examples of user interactions, agent responses, user command templates, task evaluators, etc.**
>
> Thank you for this very important feedback. We have added the corresponding examples into our rebuttal revision's appendix A.2.
>
> ### **Q1. About the "astronomy" tasks:**
> **Clarification on Tasks Selection**:
>
> The task domains in our model were chosen based on historical data concerning the most common inquiries received by our Agents. These domains include:
> 1. `Data`: Involves tasks such as data retrieval, processing, and analysis.
> 2. `Tools`: Focuses on utilizing tools to perform actions or solve tasks.
> 3. `Entertainment`: Covers tasks related to finding information on entertainment options such as movies, or executing entertainment-related actions, like playing games.
> 4. `Sciences`: Originally included specialized topics about sciences
>
> Right now, there is one task under sciences, which is `Finding relevant news about an astronomy problem`, and that was the reason why we were naming it “Astronomy”. However, given that this task represents just 1 of 30 human-crafted tasks, it does not markedly shift our overall focus towards astronomy. Nonetheless, we agree that this would cause confusion, and we think having a generic “Sciences” category suits better.
>
> We have updated our visualization for Human crafted Abstract Tasks in Figur 2(a) to avoid the confusion. Here is our detail task breakdown:
>
> `data: 20 tasks — 66.7%, sciences: 1 task — 3.3%, entertainment: 4 tasks — 13.3%, tools: 5 tasks — 16.7%`.
>
> Thank you again for your insightful feedback. This dialogue helps us make necessary adjustments and improve our manuscript. In future work, we are also continuously expanding our task collection to better represent wide areas of interest, reflecting the diversity and needs of our user base.

---

> ### Author Response · Authors · 2024-11-21
> **Rebuttal by Authors (part 2)**
>
> Below are the last part of our response.
>
> ### **Q2. "data" and "database" meaning? Why is the "data" category such a high proportion?**
> The "data" category encompasses tasks that involve data retrieval, processing, and analysis. Some examples we have are: `Search for properties on Zillow at a given location`, `Get list of ios apps`, `Retrieve top hot posts of LeetCode Dicuss Compensation`.
>
> The “database” category, in contrast, specifically relates to tasks that involve database interactions, primarily through queries. This involves retrieving, inserting, and managing data within a structured query language (SQL) framework or similar environments. Some examples we have are: `Executing a SQL query`, `Convert SQL to MongoDB`.
>
> The reason why the "data" category represents such a significant proportion of tasks is due to the extensive availability and utility of data-related tools provided by the RapidAPI marketplace.
>
> ### **Q3. Why DPO for preference optimization instead of other options?**
> Our paper highlights the LAM Simulator framework's ability to generate high-quality feedback during interactions between a user and an agent. This feedback provides valuable rewards, facilitating the refinement of subsequent learning algorithms.
>
> Given our focus on framework over individual algorithms, we adopt the most established training methodologies suitable for each type of base model:
>
> * For models like Large Action Models (Large Language Models with agentic capabilities), we chose Direct Policy Optimization (DPO) due to its popularity and relevance.
> * For other Large Language Models with minimal agentic capabilities, Supervised Fine-Tuning (SFT) is utilized to introduce aspects of agency.
>
> Additionally, we are exploring more nuanced algorithms for agency-specific tasks and plan to discuss these advancements in future publications.
>
> ### **Q4. What is U. Tool, U. Cat etc in the header of table 1?**
>
> In Table 1, Table 2, Table 4, Table 5, U represents “Unseen”. These tables present benchmark results from the ToolEval datasets[2] for different combinations of seen and unseen categories and tools:
>
> 1. U.Inst (Unseen Instruction & Seen Tools): this set contains tasks involving **unseen** instruction (or user command query) with tools **seen** during training phase
> 2. U.Tools (Unseen Tools & Seen Categories): this set contains tasks involving only **unseen** tools from **seen** task category
> 3. U. Tools & U.Cat (Unseen Tools & Unseen Categories): this set contains tasks of **unseen** tools from **unseen** task category
>
> ### **Q5. Table 2 shows average errors, but these numbers are meaningless without knowing the number of times the model was evaluated. Moreover, there are no standard deviation metrics in any tables and specifically table 2, it is difficult to know if the results are really statistically significant or if there is some lucky prompting changes/randomness in the inference models or evaluation LLMs.**
>
> When running inferences, we used greedy sampling strategy to remove randomness in the decoding and give consistent output across different runs. We standardized the prompt templates as the official base models (xLAM[1]  for LAM-Sim-7B and LAM-Sim-8x7B; Mixtral[3] for LAM-Sim-8x7B-Mixtral) to achieve best performance.
>
> For evaluation, we used majority vote on 5 judgements of GPT-4-0125-preview. We utilized the official ToolEval’s prompt template on GPT-4-0125-preview to determine the validity of the generated responses. We instructed the evaluator model to produce 5 separate judgement for the model response of each test instance, after which a majority voting mechanism was applied to reach a final judgment about each response's pass/fail status. This strategy helps mitigate variations and bias that single predictions might introduce, increasing the reliability of our reported results.
>
> Due to the nature of greedy sampling, where repeating runs guarantee the same output responses, and because we did a majority vote out of 5 evaluations from GPT-4-0125-preview, we did not add the standard deviation metrics into the table. In response, we can add extra clarification about the test sets we used. Each of the 3 test sets contains 200 instances, providing a robust basis for the reported averages.
>
> ### **References**
>
> [1] Zhang, J., Lan, T., Zhu, M., Liu, Z., Hoang, T., Kokane, S., Yao, W., Tan, J., Prabhakar, A., Chen, H., Liu, Z., Feng, Y., Awalgaonkar, T., Murthy, R., Hu, E., Chen, Z., Xu, R., Niebles, J. C., Heinecke, S., . . . Xiong, C. (2024, September 5). XLAM: a family of large action models to empower AI agent systems. arXiv.org. https://arxiv.org/abs/2409.03215
>
> [2] Qin, Y., Liang, S., Ye, Y., Zhu, K., Yan, L., Lu, Y., Lin, Y., Cong, X., Tang, X., Qian, B., Zhao, S., Hong, L., Tian, R., Xie, R., Zhou, J., Gerstein, M., Li, D., Liu, Z., & Sun, M. (2023, July 31). ToolLLM: Facilitating large language models to master 16000+ real-world APIs. arXiv.org. https://arxiv.org/abs/2307.16789

---

> ### Author Response · Authors · 2024-11-26
> **Did our revision and discussions meet your expectation?**
>
> Dear Reviewer 4LXX,
>
> We want to express our deep gratitude for your detailed and extremely helpful review on our works, and we have diligently worked to address your comments and concerns.
>
> Since the discussion period is ending, we wish to hear back from you to see that if our responses resolved your concerns or any further comments you have on our work. We sincerely hope our responses meet your expectations and would be really grateful if you would consider our work as an important step towards improving autonomous language agents, especially in a self-exploration setting.
>
> Once again, thank you so much for all of your invaluable feedback to our paper.
>
> With best regards,
> Authors of submission 13418

---

> > ### Comment · Reviewer_4LXX · 2024-11-29
> > **Reviewer Response**
> >
> > Thanks for the clarifications. I raised my score to a 6 to reflect that I now at least find the paper to be in a much more presentable format that is understandable to someone who is not in this exact subfield. I find it difficult to raise any higher however but my confidence of 2 should reflect that my review score should not be weighed very highly, this topic is some distance apart from what I study and research.
> >
> > Apologies for the late review!

---

> > > ### Author Response · Authors · 2024-12-03
> > > **Thank you for your feedback!**
> > >
> > > Thank you so much for reviewing our paper and raising the score! We truly appreciate your support and are committed to further refining our work.
> > >
> > > With best regards,
> > >
> > > Authors of submission 13418

---

### Author Response · Authors · 2024-11-23
**Regarding the modifications in the revised rebuttal**

We extend our sincere thanks to all reviewers for their insightful and detailed feedback. We have made the following key revisions to our manuscript based on all of the valuable comments we have received:

- Per Reviewer 4LXX's suggestions, we have enriched Appendix A.2 with examples of our core components, including: `Abstract Task, Tools Collection Documentation, Content Dataset, User Command, Input to Agent LLM, Agent Response,` and `Response from Environment`. Additionally, we have refined the visualization in Figure 2(a) to clarify the breakdown of categories.

- In response to Reviewer nYeX, we have added Table 0 to compare our framework with popular related framework on AI Agents development. We selected the number '0' to ensure clarity and consistency throughout our document during this rebuttal period, thus avoiding any potential confusion among reviewers regarding the indexing of previously existing tables. Furthermore, we have expanded Appendix A.3 to provide additional details about the process of constructing the tools collection in our framework.

- We have also made some minor textual revisions to adhere to the page limit, while adding new content in response to reviewer suggestions.

We believe these revisions have addressed the reviewers' concerns and have significantly enhanced both the quality and clarity of our manuscript. Once again, we appreciate all your thoughtful feedback.

---

### Meta-Review · Area_Chair_B9Yi · 2024-12-19

**Metareview:**

The paper introduces the LAM Simulator, a framework for enhancing Large Action Model (LAM) training through online exploration and simulated feedback. The discussion points regarding this paper are readability, experimental rigor, and the adequacy of comparisons. Although the authors provided detailed responses during the rebuttal period and supplemented with a large number of baseline experiments, it seems they did not receive clear affirmation from the reviewers. In summary, AC leans towards rejecting this paper, hoping that the authors will provide more detailed and substantial experimental supplements to improve its overall quality.

**Additional Comments On Reviewer Discussion:**

Despite the authors receiving positive scores (666), no one gave explicit support. Moreover, one of the reviewers gave a confidence score of only 2 for their score of 6, explicitly stating that this score was due to the authors' improvements in readability. The other two reviewers responded to the author's rebuttal and considered a score of 6 to be appropriate, but did not provide more explicit support.

---

### Decision · Program_Chairs · 2025-01-22

Reject